# Optimizing Antiviral Dosing for HSV and CMV Treatment in Immunocompromised Patients

**DOI:** 10.3390/pharmaceutics15010163

**Published:** 2023-01-03

**Authors:** Daan W. Huntjens, Jacob A. Dijkstra, Lisanne N. Verwiel, Mirjam Slijkhuis, Paul Elbers, Matthijs R. A. Welkers, Agnes I. Veldkamp, Marianne A. Kuijvenhoven, David C. de Leeuw, Heshu Abdullah-Koolmees, Maria T. Kuipers, Imke H. Bartelink

**Affiliations:** 1Pharmacy & Clinical Pharmacology, Amsterdam University Medical Center, Vrije Universiteit Amsterdam, 1081 HV Amsterdam, The Netherlands; 2Department of Intensive Care Medicine, Laboratory for Critical Care Computational Intelligence (LCCI), Amsterdam Medical Data Science (AMDS), Amsterdam Cardiovascular Science (ACS), Amsterdam Institute for Infection and Immunity (AII), Amsterdam University Medical Centre, Vrije Universiteit Amsterdam, 1081 HV Amsterdam, The Netherlands; 3Medical Microbiology and Infection Prevention, Amsterdam University Medical Center, University of Amsterdam, 1105 AZ Amsterdam, The Netherlands; 4Hematology, Amsterdam University Medical Center, Vrije Universiteit Amsterdam, 1081 HV Amsterdam, The Netherlands; 5Pharmacoepidemiology and Clinical Pharmacology, Faculty of Science, Utrecht Institute for Pharmaceutical Sciences, Utrecht University, Postbus 85500, 3508 GA Utrecht, The Netherlands; 6Clinical Pharmacy, University Medical Center Utrecht, 3584 CX Utrecht, The Netherlands; 7Cancer Center Amsterdam, 1081HV Amsterdam, The Netherlands

**Keywords:** antiviral agents, therapeutic drug monitoring, immune deficiency, allogeneic transplantation, herpesvirus 1, human, herpesvirus 2, human, cytomegalovirus

## Abstract

Herpes simplex virus (HSV) and cytomegalovirus (CMV) are DNA viruses that are common among humans. Severely immunocompromised patients are at increased risk of developing HSV or CMV disease due to a weakened immune system. Antiviral therapy can be challenging because these drugs have a narrow therapeutic window and show significant pharmacokinetic variability. Above that, immunocompromised patients have various comorbidities like impaired renal function and are exposed to polypharmacy. This scoping review discusses the current pharmacokinetic (PK) and pharmacodynamic (PD) knowledge of antiviral drugs for HSV and CMV treatment in immunocompromised patients. HSV and CMV treatment guidelines are discussed, and multiple treatment interventions are proposed: early detection of drug resistance; optimization of dose to target concentration by therapeutic drug monitoring (TDM) of nucleoside analogs; the introduction of new antiviral drugs; alternation between compounds with different toxicity profiles; and combinations of synergistic antiviral drugs. This research will also serve as guidance for future research, which should focus on prospective evaluation of the benefit of each of these interventions in randomized controlled trials.

## 1. Introduction

Severely immunocompromised patients, especially those who have received allogeneic hematopoietic cell transplantation (allo-HCT), are at increased risk for viral infections like herpes simplex virus (HSV) and cytomegalovirus (CMV). There are two types of HSV, HSV-1 and HSV-2, where HSV-1 mainly causes oral herpes infections and HSV-2 causes genital herpes [1]. Around 50% and 70% of the adults have been infected with CMV or HSV-1 by age 40–50, respectively [2]. The incidence of HSV-2 is lower, yet also significant with an estimated 491 billion people infected aged between 15–49 years. The reported incidence of viral infections in allo-HSCT recipients is inconsistent; however, the incidence of HSV and CMV infections has decreased in recent years due to effective preemptive therapies and prophylaxis [3]. Nevertheless, prolonged immunosuppression makes allo-HCT recipients vulnerable to these viral infections (e.g., reactivation of latent HSV or CMV) and occasionally results in fatal complications [3].

HSV is a common and contagious virus, and most adults and a large proportion of pediatric patients are HSV seropositive [4]. Latent herpesvirus can be reactivated after HCT, causing typical mucocutaneous lesions. Chronic mucocutaneous lesions may extend into deeper cutaneous layers, leading to easy friability and tissue necrosis [5]. Moreover, HSV reactivation can lead to HSV pneumonia or HSV encephalitis. HSV seropositive patients undergoing allo-CT should receive drug prophylaxis.

CMV disease arises from a CMV infection and can be defined as a primary or recurrent infection. CMV infection and disease are important causes of morbidity and mortality in immunocompromised patients, such as allo-HCT patients [6]. CMV disease can lead to various clinical symptoms like pneumonia, gastrointestinal disease, hepatitis, and retinitis. In case of a high CMV load, allo-HCT patients may receive pre-emptive therapy with (val)ganciclovir or foscarnet. Prophylactic therapy for CMV infection with letermovir is limited to allo-HCT patients at increased risk for mortality.

Antiviral therapy to treat CMV and HSV infection/disease is challenging for various reasons. Allo-HCT patients use immunosuppressive drugs such as ciclosporin or tacrolimus, which may cause renal toxicity. Renal impairment limits antiviral treatment options, as an accumulation of antivirals may further enhance renal toxicity. Furthermore, the impaired immune status of these patients may result in the need for prolonged antiviral treatment, even beyond the licensed treatment period. Prolonged antiviral therapy, combined with ongoing viral replication, may lead to the selection of resistant viral strains and (mainly when using valganciclovir <100 days post-allo-HCT) may cause myelosuppression and grade III/IV cytopenia [7].

In Box 1, we describe two patients from our medical center demonstrating the difficulty of treating viral reactivation in immunocompromised patients. Antiviral drugs have a narrow therapeutic window, show significant pharmacokinetic variability, and for some viral infections, there is evidence of an association between high plasma concentrations and risk of toxicity or low plasma concentrations and reduced efficacy. This scoping review aims to present the current pharmacokinetic (PK) and pharmacodynamic (PD) knowledge of antiviral drugs in immunocompromised patients. This knowledge supports healthcare practitioners when providing pharmacologically based optimization strategies for CMV/HSV treatment and suggesting individualized treatment options to improve clinical outcomes.

Box 1Two patient cases illustrating the difficulties of treating HSV in immunocompromised patients  **Case 1:** A 37-year-old woman was referred to our hospital with refractory acute myeloid leukemia. She underwent a 10/10 HLA matched unrelated-donor peripheral blood hematopoietic cell transplantation (HCT) after a FLAMSA-RIC regimen (fludarabine, amsacrine, and cytarabine, combined with reduced intensity conditioning with 4-Gy total body irradiation, high-dose cyclophosphamide and antithymocyte globulin (ATG)). The recipient HSV serostatus was positive, and the donor was not tested. She used valacyclovir as HSV prophylaxis (500 mg twice daily (BID)). Despite sirolimus and mycophenolate mofetil immunosuppression, grade 4 acute intestinal graft versus host disease (aGvHD) developed for which she received several lines of treatment: intravenous prednisolone (2 mg/kg) followed by ruxolitinib which was later combined with vedolizumab. During this treatment, she developed HSV-1 oral mucositis for which intravenous (IV) acyclovir 10 mg/kg three times daily (TID) was started, as shown in the Figure 1. Additional examination showed an HSV load of 59,000 copies/mL in peripheral blood. After 1 week of acyclovir treatment, there was no clinical response. Moreover, her HSV load in peripheral blood increased towards >100,000 copies/mL. Replacement of acyclovir with foscarnet (BID 60 mg/kg) successfully improved her oral ulcers and the HSV load in blood became undetectable after 2 weeks of treatment. Prophylactic valacyclovir (500 mg BID) was restarted. However, 5 days later her oral ulcerations relapsed and were again HSV-1 positive, treatment was switched to foscarnet (40 mg/kg TID). Her eGFR (estimated using CKD-EPI) was >90 mL/min/1.73 m^2^ at that timepoint. We decided to continue treatment until full recovery of all oral ulcerations as she was still immunocompromised due to her post allo-HCT status and continuous treatment for aGvHD. Unfortunately, after four weeks of foscarnet treatment her oral ulcerations were minimal but still present and HSV-1 positive. Moreover, the first signs of renal toxicity (eGFR ranging between 68–90 mL/min in this period) were observed. Doses were adjusted according to the SmpC. Analysis for susceptibility to antiviral agents was performed. We found a mutation in UL30 A605V which is associated with acyclovir and foscarnet resistance. No mutations were found in thymidine kinase. We requested access to brincidofovir under an emergency IND application, which was rejected. Cidofovir treatment was initiated at 5 mg/kg IV weekly for 2 weeks followed by once every 2 weeks. The baseline eGFR was 90 mg/mL She received 2 doses of cidofovir. Although her oral ulcerations responded well, her clinical condition worsened (WHO 3, losing weight, nausea) and her kidney function deteriorated within three weeks to 42 mL/min/1.73 m^2^. Cidofovir was prematurely stopped and acyclovir prophylaxis was restarted. In the weeks thereafter her clinical condition improved slowly and kidney function increased to approximately 80 mL/min/1.73 m^2^. Unfortunately, 5 weeks later, she developed new HSV-1 positive oral lesions. Via an Early Access Program, we received pritelivir. After 1 week of treatment her oral ulcerations resolved, and she completed 4 weeks of treatment without significant toxicities. Thereafter valacyclovir prophylaxis was restarted.  **Case 2:** Female 61 years of age, underwent an upfront allo-HCT for chronic myelomonocytic leukemia (CMML) after a FLAMSA-RIC regimen. Shortly after the transplant (day + 6) she developed HSV positive orolabial lesions, for which acyclovir was started Figure 2. Additional examination showed an HSV load of 59,000 copies/mL in peripheral blood. After 14 days of acyclovir treatment, the oral labial HSV lesions appeared clinically refractory and treatment was switched to foscarnet. During foscarnet therapy, her renal function deteriorated from 64 to 32 mL/min/1.73 m^2^ and doses were adjusted according to the SmpC. Moreover, under continuous treatment with foscarnet, our patient developed pneumonitis. Bronchoalveolar lavage (BAL) fluid was negative for bacteria, pneumocystis, fungi and respiratory viruses except for low detectable HSV-1 (1600 copies/mL). The diagnosis of HSV pneumonitis was initially discarded as the HSV-1 load in the peripheral blood was undetectable and her orolabial lesions showed signs of improvement. Therefore, treatment with corticosteroids was initiated to treat a possible drug-induced pulmonary toxicity. In the following days, she developed respiratory failure and was transferred to the Intensive Care Unit (ICU). BAL was repeated and showed a strongly increased HSV load (>100,000 copies/mL) indicating a clinically refractory and progressive HSV pneumonitis under foscarnet. Acyclovir was briefly added with the hypothesis of two HSV clones with different susceptibility; the orolabial clone being foscarnet sensitive and acyclovir resistant, the pulmonary clone acyclovir sensitive and foscarnet resistant. Analysis for susceptibility to antiviral agents showed mutations in HSV polymerase associated with acyclovir and foscarnet resistance. Here, pritelivir was started via the Early Access Program and both acyclovir and foscarnet were halted. Her HSV responded well to the treatment. After 4 weeks of treatment, HSV was repeatedly undetectable in BAL. Unfortunately, the patient developed an idiopathic lung syndrome and multiorgan failure and died while on pritelivir treatment.

**Scheme 1 pharmaceutics-15-00163-sch001:**
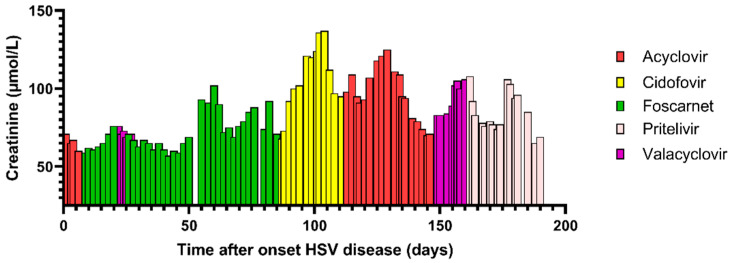
Creatinin serum levels (μmol/L) over time (case 1).

**Scheme 2 pharmaceutics-15-00163-sch002:**
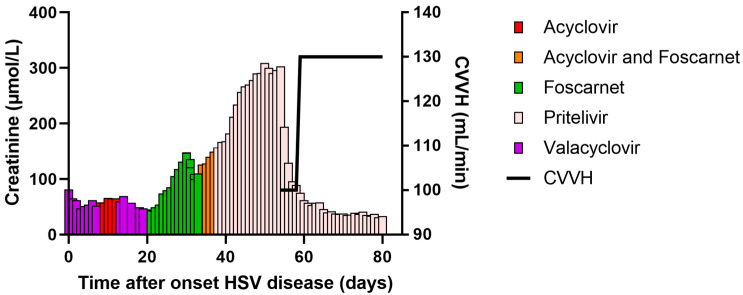
Creatinin serum levels (μmol/L) over time (case 2).

## 2. Acyclovir

### 2.1. Background

Acyclovir is a nucleoside analog that is used in the treatment of herpes virus infections, including HSV and varicella-zoster virus infections [8].

Valacyclovir, the L-valyl ester of acyclovir, is the prodrug of acyclovir. Acyclovir is available as an oral, topical, and intravenous formulation; the prodrug valacyclovir is available as an oral formulation [9].

### 2.2. Mechanism of Action

Valacyclovir is converted to acyclovir, which is subsequently converted to acyclovir monophosphate by viral thymidine kinase. Cellular kinases convert acyclovir monophosphate into acyclovir triphosphate, which is responsible for its antiviral activity. Intracellularly, the activated nucleoside competes with deoxyguanosine triphosphate for DNA incorporation by viral DNA polymerases, leading to DNA chain termination and disruption of viral DNA synthesis [8,10]. (Figure 1)

Acyclovir resistance in HSV strains is mainly reported in immunocompromised patients and has been linked to mutations in or deficiency of viral thymidine kinase in most cases. Long-term prophylaxis or treatment with acyclovir is a risk factor for the development of acyclovir resistance [11,12,13].

### 2.3. Activity against HSV and CMV

In vitro data suggest that acyclovir is effective against HSV-1 and HSV-2 and CMV [8,14] (Table 1).

In vivo, the activity of acyclovir against HSV-1 and HSV-2 was confirmed, but the drug showed moderate effectiveness for CMV in the clinic. This might be explained by the mechanism of activation: the triphosphate form of acyclovir is formed through phosphorylation of acyclovir in the presence of virus-specific thymidine kinase. CMV does not have this virus-specified thymidine kinase. For CMV, the UL-97 protein kinase seems to be responsible for the phosphorylation of acyclovir [15]. Therefore, (val)acyclovir may be effective as prophylaxis of CMV, but relatively high acyclovir concentrations are required. Therefore, acyclovir is inferior to ganciclovir in preventing CMV disease [16]. Furthermore, the drug cannot be used in the treatment of CMV [8,14].

### 2.4. Pharmacokinetics

Acyclovir has poor oral bioavailability (15–30%). Valacyclovir is nearly completely absorbed after oral administration and rapidly converted to acyclovir and L-valine. The bioavailability of acyclovir from valacyclovir is 54%. Peak plasma concentration is reached after 1–2 h. Acyclovir’s protein binding is low (9–33%) and its volume of distribution approximates 26 L. Acyclovir concentrations in cerebrospinal fluid are approximately 50% lower than those in plasma. Acyclovir is mainly excreted in the urine, mostly unchanged, but partly as its main metabolite 9-Carboxymethoxymethylguanine (CMMG, 10–15%). However, in patients with impaired renal function, a larger proportion of acyclovir is excreted as CMMG. Clearance is mainly via renal elimination and encompasses glomerular filtration and tubular secretion. Acyclovir’s plasma half-life is about 2.9 h and it increases to 19.5 h in patients with chronic kidney disease [8,9,11,17].

### 2.5. Pharmacodynamics

Data on acyclovir exposure in relation to clinical efficacy and toxicity are limited.

The PK/PD index that seems most suitable for acyclovir is the time that the plasma concentration remains above the EC_50_ (Table 1) [18,19].

As only unbound concentrations can reach the target site, therapeutic drug monitoring of Ctrough values, aiming for an unbound concentration above EC_50_ may benefit treatment outcomes. Total acyclovir or CMMG plasma/CSF concentrations may be related to neurological toxicity [17,20]. TDM of acyclovir might benefit immunocompromised patients, although limited data are available.

### 2.6. Side Effects

An important side effect of acyclovir is acute renal failure. Acyclovir-induced renal failure may be caused by intratubular deposition of acyclovir crystals in the kidney due to the low solubility of acyclovir in urine. The risk of deposition of acyclovir crystals in the kidney can be avoided by slow drug infusion over 1 to 2 h and volume repletion. Acyclovir-induced renal injury without acyclovir crystal obstruction is also described [21,22]. Another important side effect of acyclovir is neurological toxicity, mainly in patients with renal failure. This might be caused by the accumulation of the CMMG metabolite of acyclovir. (see Pharmacokinetics). CMMG has been detected in the cerebrospinal fluid (CSF) of acyclovir-treated patients [20]. The mechanism of CMMG-induced neurotoxicity is unknown. Acyclovir’s moderate to severe side effects, including neurotoxicity, may be associated with high serum acyclovir levels (>25 mg/L). Moreover, detectable CMMG levels in CSF are associated with neurotoxicity [11,23,24,25].

**Table 1 pharmaceutics-15-00163-t001:** Comparison of effective concentration in vitro to the observed concentration in plasma at standard dosing and potentially toxic concentration and proposed total plasma target range. The following variables were taken into account for the proposed total plasma target range (mg/L): mean observed total Css, toxic total concentration, in vitro EC_50_ concentrations for HSV and CMV, and the percentage of plasma protein binding. It is recommended to use model informed precision dosing (MIPD) when sampling C_max_ at T_max_.

Drug	EC_50_ HSV (μM) [18]	EC_50_ CMV (μM) [18]	Mean Observed Total C_ss_ (μM)	Mean Predicted Unbound C_ss_ (μM)	Toxic Total Concentration (μM)	Proposed Total Plasma Target Range (mg/L)
Acyclovir	0.1–20	n.a. ^$^	8.9	6.8–9.3	20.0	C_trough_ 1.0–5.0
Ganciclovir	0.2–2.5	0.1–37	7.8	7.8	15.6–47	C_trough_ 2.0–4.0
Foscarnet	50–250	27–300	150	135	Unknown	C_trough_ > 20.0
Cidofovir	0.6–9	0.6–9.5	41 ^#^ [26]	41 ^#^ [26]	Unknown	C_max_ > 11.5
Letermovir	n.a.^$^	7–61 * 10^−4^ [27]	4.5–38 ^#^ [27]	4.45–38 * 10^−2 #^	38 ^#^	C_max_ 2.5–22.0
Pritelivir	2.6–2.9 * 10^−2^ [28]	n.a. ^$^	1.66 [29]	0.04 [29]	Unknown	C_trough_ 0.83–5.0
Brincidovovir	0.9–2.9 * 10^−2^	0.1–3 * 10^−2^	0.85 ^#^ [30]	8.5 * 10^−3 #^	Unknown	C_max_ > 0.48
Maribavir	Inactive	0.08–0.32	44	0.66	Unknown	C_max_ > 15

* Estimation is based on the portion which is protein bound. ^$^ Acyclovir and pritelivir are not recommended as therapy for CMV, letermovir is not recommended for therapy of HSV. ^#^ Concentration refers to the maximum blood concentration (C_max_) C_ss_ = plasma C_trough_ at steady state at standard dosing, unless stated otherwise.

## 3. Ganciclovir

### 3.1. Background

Ganciclovir is another nucleoside analog of guanosine with antiviral activity against CMV and HSV. It is currently the recommended first-line option for the prevention and treatment of CMV infection and disease in solid organ transplant (SOT) and stem cell transplant recipients [31,32,33,34].

### 3.2. Mechanism of Action

Ganciclovir is a synthetic acyclic nucleoside analog of guanosine. In cells infected with HSV, the viral thymidine kinase phosphorylates ganciclovir to the monophosphate form, and cellular enzymes convert the monophosphate to ganciclovir triphosphate similar to the mechanism of action (MOA) of acyclovir in HSV (Figure 1) [35].

Unlike acyclovir, ganciclovir is phosphorylated in CMV infected cells as well. UL97 viral kinase can phosphorylate ganciclovir to ganciclovir monophosphate, where cellular kinases form ganciclovir triphosphate [35,36,37,38,39]. Ganciclovir triphosphate competitively inhibits DNA synthesis, which is catalyzed by the viral DNA polymerase UL54, resulting in a slower viral DNA chain elongation [40].

### 3.3. Activity against HSV and CMV

Ganciclovir is the first antiviral drug to be effective in the treatment of CMV disease in humans. In tissue culture, ganciclovir exhibits excellent antiviral activity against CMV, HSV type 1 and 2, Epstein–Barr virus, varicella-zoster virus and, human herpesvirus 6 [35]. In vitro studies demonstrated the antiviral efficacy of ganciclovir against CMV to be 8-20 fold greater than that of acyclovir and ganciclovir tends to be as active against HSV type 1 and 2 as acyclovir (Table 1) [41]. In general, ganciclovir is considerably more effective than acyclovir and equally effective as foscarnet against CMV [42,43]. Although ganciclovir has major advances in CMV disease management, its extended use is limited by the risk of drug resistance. Resistance to ganciclovir arises from mutations in the UL97 and UL54 genes. Proposed risk factors for the development of ganciclovir resistance include inadequate plasma or tissue drug concentrations, frequent discontinuation of treatment, longer duration of antiviral use, the use of potent immunosuppressive regiments, and a CMV donor positive/recipient negative serostatus [44,45].

### 3.4. Pharmacokinetics

Given ganciclovir’s low oral bioavailability (~6%), the prodrug valganciclovir was developed for oral treatment with an improved oral bioavailability (~60%) [32,38]. The bioavailability of valganciclovir is 24–56% higher in the fed condition than that in the fasted condition. After absorption, the prodrug valganciclovir is rapidly hydrolyzed into ganciclovir. The time to maximum concentration (Tmax) of ganciclovir after valganciclovir intake is 1.0–3.5 h [31,38].

The plasma protein binding of ganciclovir is negligible (1–2%), and the volume of distribution is 0.59–0.89 L/kg. Ganciclovir is eliminated mainly through the kidneys by glomerular filtration and active tubular secretion. 90% of ganciclovir is excreted in the urine unchanged. Ganciclovir has an elimination half-life of 2.6–4.4 h. In patients with mild renal impairment, ganciclovir clearance is almost half of the clearance value in healthy subjects [38,46].

Considerable pharmacokinetic variability was observed in immunocompromised patients after the administration of intravenous ganciclovir or oral valganciclovir. AUC coefficient of variation values ranged from 27 to 55% in HCT patients, 6% to 53% in SOT patients, 12.9% to 31% in HIV patients, and 17 to 20% in healthy subjects. In addition to the observed significant interpatient pharmacokinetic variability, several studies reported a higher AUC and longer half-life in HCT patients treated with either ganciclovir or valganciclovir compared with other populations previously studied. The suggested hypotheses are alterations in renal tubular secretion and increased bioavailability of valganciclovir in the HCT population, with gut toxicity from conditioning chemotherapy regimens being theorized to be responsible for the latter [31].

### 3.5. Pharmacodynamics

In a systematic review studying the clinical pharmacokinetics, pharmacodynamics, and toxicodynamics of ganciclovir in HCT-recipients, an association between elevated ganciclovir peak and trough concentrations and neutropenia was found in only one out of eight studies. A ganciclovir dose >12 mg/kg/day was identified as a risk factor for ganciclovir-induced thrombocytopenia in a singular study. No association between serum ganciclovir peak and/or trough concentrations and less common adverse effects of ganciclovir, such as nephrotoxicity, was observed [31,38,47]. However, multiple studies apply a lower limit based on IC_50_ values of approximately 2 mg/L, and an upper limit of 4 to 12 mg/L, based on an increased risk of toxicity at higher dose levels, combined with an observed large variability in PK among patients [48].

### 3.6. Side Effects

Ganciclovir can cause neutropenia, thrombocytopenia, leukopenia, and elevated serum creatinine. The rate of myelotoxicity varies between specific patient groups. In HCT-recipients, myelotoxicity rates are higher compared to heart-transplant recipients, with neutropenia and thrombocytopenia occurring in 41% versus 7% and 57% versus 8%, respectively [49].

A high incidence of impaired renal function has been observed in transplant patients receiving intravenous ganciclovir, with elevated serum creatinine values observed in 43% to 58% of patients enrolled in these trials. However, nephrotoxicity is reversible in most instances when ganciclovir is withdrawn. Reduced renal function at treatment onset, which frequently occurs in transplant patients as a result from the use of the calcineurin inhibitors cyclosporine and tacrolimus, has been identified as a risk factor for ganciclovir-induced toxicity [31,49,50,51].

## 4. Foscarnet

### 4.1. Background

Foscarnet was synthesized in 1924, yet its antiviral activity was found much later in 1978. Foscarnet is an inhibitor of DNA polymerase and HIV reverse transcriptase, with broad antiviral activity against CMV, HSV, and HIV. [52] Since oral bioavailability is limited, the drug can only be administered by intravenous infusions, limiting its use to in-hospital settings [53].

Foscarnet therapy consists of an induction phase with higher doses of foscarnet, followed by a maintenance phase with lower dosages. This complex dosing schedule prevents a fast reactivation of CMV after ending the induction phase [54].

Previous foscarnet studies in immunocompromised patients are limited to CMV retinitis in patients with AIDS or the treatment of CMV infections in stem cell recipients. Studies on pharmacokinetics and pharmacodynamics of foscarnet are scarce.

### 4.2. Mechanism of Action

Foscarnet inhibits viral DNA polymerase of HSV, CMV, varicella-zoster virus, and Epstein–Barr virus. Inhibition of DNA polymerase is induced by preventing pyrophosphate exchange in the DNA replication process and thereby inhibiting viral proliferation (Figure 1) [55].

### 4.3. Activity against HSV and CMV

Foscarnet is a pyrophosphate analog, active against HSV and CMV (Table 1), including strains with acyclovir or ganciclovir resistance. Most efficacy studies are comparing foscarnet and ganciclovir, showing similar treatment results (see ganciclovir section). Notably, the result of a randomized controlled trial in 213 patients post allo-HCT, preemptive therapy of CMV infection with foscarnet was similarly effective as ganciclovir, but associated with a lower proportion of patients who develop severe neutropenia and who required discontinuation of antiviral therapy due to hematotoxicity [42].

It seems that treatment with foscarnet should be continued after induction therapy. One study in immunocompromised patients with HIV showed that a 14-day course of foscarnet reduces CMV-loads significantly. However, after discontinuation of the drug, there was a rapid CMV-reactivation observed. The authors suggest that maintenance therapy was required for successful CMV treatment [54].

Foscarnet can also be used as a prophylactic treatment in specific immunocompromised patients. In a relatively small patient group of 57 patients post HCT, foscarnet prophylactic treatment (90 mg/kg once daily from day 7–27 after transplantation) seemed to reduce HSV-reactivation, but failed to prevent cases of HSV encephalitis [56].

In patients unable to receive ganciclovir therapy, due to the myelosuppressive side effects of ganciclovir, foscarnet was successfully used to prevent CMV disease. A relatively low dose of 60 mg/kg once daily (QD) was used in 39 patients. Of these 39 patients, six patients (15%) eventually developed an active CMV infection, requiring further antiviral treatment [57]. It should be noted that in both studies, the rationale for the prophylactic dosing schedule of foscarnet was missing.

### 4.4. Pharmacokinetics

The volume of distribution of foscarnet at steady state is 0.4–0.6 L/kg. Protein binding is low (<20%). The concentration time curve of foscarnet shows a tri-exponential decay with respective half-lifes of 0.45, 3.3, and 18 h [58]. Foscarnet is eliminated by the kidneys mainly through glomerular filtration, and dose adjustments seem necessary to prevent additional nephrotoxicity. The product leaflet of foscarnet provides recommendations to reduce the dose according to the obsolete Cockcroft & Gault estimation of the glomerular filtration rate (GFR). Prior studies show that the CG equation provides a systematic overprediction of the renal function and subsequent foscarnet dosing in patients <65 years, specifically in obese patients, and underpredicts in elderly and cachexic patients [59,60]. A more adequate measurement of GFR such as using iothalamate or TDM may help to optimize the dose in obese or cachexic patients with renal dysfunction. Only limited data is available on the PK of foscarnet in immunocompromised patients. In a comparison of two dosage regimes of foscarnet, there was a large interindividual variation observed in foscarnet C_max_ and AUC [61]. This large variation was confirmed in a trial that included 31 patients with foscarnet therapy for CMV retinitis in HIV-positive patients, with a ten-fold interindividual variation in AUC, even after adjusting the dose based on the renal function [62].

### 4.5. Pharmacodynamics

There is some evidence that the foscarnet AUC relates to treatment response. In a cohort of 29 HIV-positive patients with CMV-retinitis, AUC was associated with the time to progression [62]. In addition, in a study comparing two prophylactic dosing schemes of foscarnet in 20 patients after allogenic bone marrow transplantation, a dose-dependent prophylactic effect was observed on blood cultures positive for CMV, CMV disease, and transplant-related mortality [63]. This dose–response relation and large interindividual variation in PK could support the hypothesis that TDM of foscarnet might be beneficial. A therapeutic window of foscarnet needs, however, is still to be established.

### 4.6. Side Effects

The major side effects of foscarnet therapy are impaired renal function, anemia, and electrolyte disturbances [52]. These side effects are also described in immunocompromised patients [64,65]. There is no relation between serum concentrations and the occurrence of side effects reported in the literature.

## 5. Cidofovir

### 5.1. Background

Cidofovir is a nucleotide analog developed for the treatment of CMV infections in patients with acquired immunodeficiency syndrome (AIDS). Randomized controlled clinical studies of efficacy have been limited to patients with AIDS and CMV retinitis [66,67]. However, cidofovir also has potent activity against other herpesviruses. It was the first nucleotide analog with broad antiviral activities [68].

Therefore, cidofovir is indicated for the treatment of immunocompromised patients with acyclovir-, ganciclovir-, and foscarnet-resistant HSV, VZV or, CMV infections.

Cidofovir is available as an intravenous formulation only.

### 5.2. Mechanism of Action

Cidofovir is an antiviral nucleotide analog of cytosine. After phosphorylation by cellular kinases from monophosphate to its active diphosphate form, it inhibits viral DNA polymerase pUL54 and incorporates itself into the viral DNA which interrupts further elongation of the virus [68]. Cidofovir resistance is caused only by DNA polymerase mutations, mainly by a UL54 mutation which also results in cross-resistance to ganciclovir.

### 5.3. Activity against HSV and CMV

Cidofovir has antiviral activity against a broad spectrum of DNA viruses. It is active in vivo against CMV and in vitro against all major herpesviruses (HSV, VZV, EBV), adenovirus, papillomavirus and, poxvirus. The activity spectrum is much broader than of other antiviral drugs. Cidofovir remains active against nucleoside-resistant strains of herpesviruses in contrast with acyclovir or ganciclovir due to intracellular activation by host cellular enzymes independent of viral infection. The efficacy of cidofovir has been retrospectively investigated in allogeneic HCT recipients with CMV infections and can be considered as a second-line treatment [69]. Additionally, cidofovir has proven efficacy against acyclovir-resistant mucocutaneous HSV infections [70].

### 5.4. Pharmacokinetics

Cidofovir had a prolonged antiviral action due to intracellular metabolites (cidofovir monophosphate, cidofovir diphosphate, and cidofovir-choline phosphate) with a long half-life of 48 h. Therefore, dosing is once a week 5 mg/kg for 2 weeks followed by 5 mg/kg every other week, which is a favorable pharmacokinetic profile compared to other antiviral drugs. PK has been investigated in HIV-infected patients with or without co-administration with probenecid. The protein binding of cidofovir is negligible (<0.5%). Cidofovir is excreted extensively as an unchanged drug in urine (80–100%) within 24 h. Pharmacokinetics differ when cidofovir is co-administrated with probenecid to protect the nephrons. When combined with probenecid, unchanged cidofovir has a terminal half-life of 2.4 +/− 0.5 h in plasma, meaning that creatinine clearance is consistent with a renal clearance of cidofovir by glomerular filtration [26,71].

### 5.5. Pharmacodynamics

At weekly dosing, the time above IC_50_ of cidofovir is relatively short. However, the true duration of action of cidofovir is dependent on the concentration of the intracellular activated metabolites (cidofovir monophosphate, cidofovir diphosphate, and cidofovir-choline phosphate). These metabolites contribute to the long-acting activity. Therefore, plasma concentrations of cidofovir (Time above IC_50_, C_max_, or AUC) do not correlate with the duration of action [26].

### 5.6. Side Effects

In clinical practice a major dose-limiting side effect of intravenous cidofovir is nephrotoxicity. Incidence rates between 25% and 33% have been reported. The concentration of cidofovir in renal cells is 100 times higher than in other tissues, which results in proximal tubular injury. Strategies to minimize cidofovir-associated toxicity consist of prehydration with normal saline and concomitant administration of probenecid. Probenecid prevents nephrotoxicity by blocking the active tubular secretion of cidofovir. Other nephrotoxic agents should be discontinued at least seven days prior to starting with cidofovir, which further limits its use, especially early after HCT when immunosuppressive agents are given. This side effect profile limits the current use of cidofovir. It is recommended to carefully select patients and monitor kidney function during and after the administration of cidofovir [72,73,74].

## 6. Letermovir

### 6.1. Background

Letermovir (LMV, AIC246) is a non-nucleoside CMV inhibitor and a 3,4 dihydro-quinazoline-4-yl-acetic acid derivative. Letermovir was approved in the USA and Canada in 2017 for CMV prophylaxis in HCT patients in IV and PO formulations [75,76,77].

### 6.2. Mechanism of Action

Letermovir is a viral terminase inhibitor. Letermovir inhibits the CMV DNA polymerase complex being pUL51, pUL56, and pUL89. These DNA polymerases are crucial for viral DNA to be processed and packaged. The viral DNA concatemers cannot be cleaved by inhibiting pUL56 and consequently, mature virions cannot be produced. Letermovir distinguishes itself by targeting pUL56 (Figure 1). Other antiviral drugs target the CMV UL54 gene, which encodes viral DNA polymerases [75,76,77,78]. Since virus mutations can also occur in CMV, this can affect letermovir effectivity in preventing CMV infections. However, mutations in CMV seem to occur mainly in pUL56 and are less common for pUL89 and pUL51.

### 6.3. Activity against HSV and CMV

Due to its MOA, letermovir is only effective as CMV prophylaxis in HCT patients, not for treatment. Letermovir fails to prevent other common viral infections such as HSV and VZV [75,76,77,78].

### 6.4. Pharmacokinetics

Letermovir has a high bioavailability of >90% in healthy volunteers while it had a moderate bioavailability of around 35% in HCT patients. Letermovir’s Cmax increases by 30% with food and its bioavailability increases to 85% with concomitant use of cyclosporine. Therefore, half of letermovir’s dose (240 mg) is indicated with the concomitant use of cyclosporine (e.g., OATP1B1/3 transporter inhibitor). Cyclosporin increases the AUC of letermovir by a factor of 2.11. Letermovir has time to peak (Tmax) around of 1.5–3 h and has a half-life of 10 h. Hepatic uptake is mediated through OATP1B1/3. A minor portion of letermovir is metabolized through UGT1A1/1A3-mediated glucuronidation. Letermovir’s efflux is mediated through P-gp and BCRP in the liver as in the intestines.

Letermovir has a high protein binding (>99%) and is mainly excreted through feces (93 of which 70% unchanged). Kidneys play a minor role in letermovir’s excretion (<2%). Furthermore, letermovir inhibits CYP2C8 and inhibits CYP3A4 moderately (e.g., increasing amiodaron plasma levels) while it induces CYP2C9 and CYP2C19 (e.g., decreasing voriconazole plasma concentration). Furthermore, rifampicin decreases letermovir plasma concentration through UGT1A1 and P-gp induction. Letermovir increases the plasma concentration of medication which are substrates of OATP1B1/3 transporters like statins [75,76,77].

### 6.5. Pharmacodynamics

For prophylactic treatment, 480 mg QD of letermovir (oral or IV) is started after HCT (between days 0–28) till 100 days post HCT. Plasma concentrations of 0.2-50 mg/L are reached. Letermovir’s median EC_50_ against CMV isolates was 2.1 nM (range: 0.7–6.1 nM, N = 74) in clinically isolated cultures (Table 1). The EPAR of letermovir reported C_max_ values between 2.5 and 22 mg/L as assessed in the phase 3 study in HCT patients. Furthermore, the concentration of 21 mg/L was set as the most conservative upper limit of C_max_ at standard dosing [27,75,76,77].

### 6.6. Side Effects

The main side effects of letermovir are uncommon and mostly affect the gastrointestinal tract, including gastritis and nausea. Furthermore, dyspnea and hepatitis can occur.

## 7. Pritelivir and Amenamevir

### 7.1. Background

Pritelivir (formerly named AIC316 and BAY 57-1293) and amenamevir (ASP2151) belong to a novel active class of non-nucleoside compounds: helicase-primase inhibitors. In 2017 amenamevir was approved in Japan (oral dosage of 400 mg for 7 days) for therapy of herpes zoster, whereas the first clinical trial results of (oral) pritelivir were presented in 2014. Pritelivir is currently under investigation for resistant and refractory HSV [79].

### 7.2. Mechanism of Action

Pritelivir and amenamevir prevent de novo synthesis of the virus through inhibition of the viral helicase-primase complex. This complex comprises three proteins, encoded by the UL5 (helicase), UL52 (primase), and UL8 (scaffold protein; shown to promote primer synthesis) genes (Figure 1) [80]. These proteins are crucial for viral DNA replication [80]. This mechanism differs from the nucleoside analogs which terminate ongoing DNA chain elongation through inhibition of viral DNA polymerase. Furthermore, their activity is not dependent on the infection having progressed to the point where specific viral thymidine kinase is synthesized. The compounds directly inhibit the replication of HSV, limit the intracellular viral load, and prevent the spread of the virus into new cells. Another advantage is that these compounds remain active against mutants that are thymidine kinase defective [28,81,82]. This difference in mechanism is promising, as ganciclovir- and acyclovir-resistant HSV infections are increasingly reported in immunocompromised patients. Additionally, a combination of acyclovir and Helicase-Primase Inhibitors demonstrated a synergistic/additive effect against acyclovir-sensitive HSV and VZV strains in vitro and in vivo [81,83]. However, as these drugs are currently intended for the treatment of acyclovir-resistant strains, the value of this synergy for the clinic is currently unknown.

### 7.3. Activity against HSV and CMV

The drugs are selective for HSV. Amenivir is also active against VZV, but pritelivir is inactive against other viruses such as CMV and VZV (Figure 1) [18,81,84].

### 7.4. Pharmacokinetics

PK of pritelivir has been demonstrated in a single and multiple-dose ascending trial. [84,85] A dose-proportional increase in exposure (linear PK) has been observed. Terminal elimination half-life varied between 60–70 h and AUC of approximately 90.8 mg*h/L after a single 100 mg dose, with 75% bioavailability [85]. The protein binding of pritelivir is high: in vitro studies predict 97.3% [85].

Amenamevir PK was analyzed in four randomized phase 1 studies. [86] The terminal elimination half-life varied between 7 and 8 h after a single dose and at steady state. Exposure was affected by food, with AUC increased by about 90%. The oral bioavailability of amenamevir is 86%, Approximately 75% of amenamevir is excreted in the feces and 20% in the urine [86].

### 7.5. Pharmacodynamics

A good concentration response has been observed for both drugs in preclinical translational studies. A single dose of 300 mg of amenamevir preserved a mean plasma concentration over nine times higher than the EC_50_ after 24 h and seven times higher than the EC_50_ in a mouse skin infection model, whereas for pritelivir preclinical experiments show similar results [29,87].

Clinical development of pritelivir is ongoing but was delayed after blood and skin-related findings in monkeys in a 39-week toxicity study. In multiple-dose escalation trials doses between 5 mg, 25 mg, and 100 mg once daily (QD); 400 mg weekly for 21 days shows that pritelivir reduced the rates of genital HSV shedding in otherwise healthy patients with genital herpes, compared to placebo in a dose-dependent manner [88]. Compared to valacyclovir, pritelivir at 100 mg QD dosing, showed a lower percentage of swabs with HSV detection (2.4% vs. 5.3%, *p*  =  0.01) [89]. Genital lesions were presented on 1.9% and 3.9% of the days in the PTV and VACV cohort, respectively [89].

Effective pritelivir therapy in recurrent genital herpes in an immune-compromised allo-SCT recipient infected with acyclovir-resistant HSV-2 has been described [90]. Studies in immunocompromised mice suggest that the efficacy of neither drug was affected by the host’s immune status in terms of effective oral dose [29,91]. A more robust antiviral response was observed using amenamevir compared to valacyclovir in these immune-deficient hosts [91].

Recruitment to a new study concerning the efficacy and safety of 100 mg QD PTV (following a 400 mg loading dose) up to a maximum of 42 days in acyclovir-resistant mucocutaneous HSV infections in immunocompromised individuals (PRIOH-1, NCT03073967) is ongoing [79]. In this study, the drug will be compared with standard doses of foscarnet. The study will show whether a 42-day treatment period of helicase-primase inhibitors is sufficient to treat active HSV lesions during immune deficiency.

### 7.6. Side Effects

It is claimed that amenamevir and pritelivir are well-tolerated but the toxicity profile is not yet thoroughly investigated and is used for a relatively short treatment period (maximum of 42 days) [80,92]. In immune-competent patients, both drugs do not cause serious side effects. Amenamevir has been used to treat at least 1,240,000 individuals with a herpes zoster diagnosis. In these groups of patients, the frequency of adverse reactions could be lower than in people treated with the other anti-herpes-virals [87]. In patients with eGFR until 30 mL/min, both drugs seem safe.

## 8. Maribavir

### 8.1. Background

Maribavir (formerly named 1263W94) is a novel benzimidazole I-riboside compound, and although it has some similarities to BDCRB (2-bromo-5,6-dichloro-1-beta-d-ribofuranosyl-1-H-benzimidazole), it acts through a different mechanism. Maribavir is an investigational drug being evaluated in transplant recipients with CMV infection. Maribavir has activity against human CMV and Epstein–Barr virus, but not to other herpesviruses.

### 8.2. Mechanism of Action

Maribavir directly inhibits UL-97 kinase, unlike ganciclovir, which needs to be phosphorylated by UL-97 kinase to become an active inhibitor of DNA polymerase [93]. The complete function of how UL-97 and its function on replication and how maribavir impacts the inhibition of viral replication remains unclear.

### 8.3. Activity against CMV

In doses of ≥400 mg BID maribavir is effective for the treatment of RR-CMV (refractory or resistant CMV) and ineffective against HSV.

### 8.4. Pharmacokinetics

Maribavir is rapidly absorbed and displayed potential non-linear pharmacokinetics. All dosing schemes had similar dose corrected-trough concentrations, but the 400 mg doses yielded a plasma peak concentration 2.5 times higher than the 100 mg dose. Furthermore, the 400 mg BID demonstrated the highest 24 h drug exposure. Cmax was achieved 1–3 h after administration with a plasma half-life of 3–5 h. Protein binding is approximately 98.5% [94]. Maribavir is extensively hepatically metabolized by CYP3A4 and primarily eliminated via biliary excretion. Clearance is not impacted by renal impairment (<3% urinary excretion). Administration with meals high in fat content decrease maribavir concentrations by 30%. Maribavir inhibits P-gp activity but did not affect CYP2D6 activity [95].

### 8.5. Pharmacodynamics

The first randomized clinical trial investigating prophylaxis therapy failed to show superiority [96]. This was probably caused by the low dose of maribavir used in this study (100 mg BID). However, a recent phase 2 study in which maribavir was used for the pre-emptive therapy after HCT show similar efficacy to valganciclovir. Furthermore, maribavir was superior to investigator-initiated therapy of (val-)ganciclovir, foscarnet, or cidofovir in refractory disease.

### 8.6. Side Effects

Maribavir has been associated with taste disturbances but causes no nephrotoxicity or hematologic toxicities. Dysgeusia is described as “metallic” or “bitter” tastes and was reported by 65% of the patients [97]. Additionally, nausea and vomiting were reported. The intensity and frequency of the taste disturbance increase with dose, but no association study of exposure and toxicity at standard dose was included.

## 9. Brincidovovir

### 9.1. Background

Brincidofovir is a lipid conjugate of cidofovir, enabling increased intracellular levels of cidofovir diphosphate (Figure 1). The EC_50_ value of brincidofovir is low compared to any of the other antivirals currently in the clinic (Table 1) [18] Lower rates of nephrotoxicity and myelotoxicity make it a favorable alternative to cidofovir treatment for smallpox and other infections such as HSV and CMV [98]. However, although licensed by FDA, currently, brincidofovir is not commercially available.

### 9.2. Pharmacokinetics

The bioavailability of brincidofovir is 13.4% with a T_max_ at 3 h. Brincidofovir is a prodrug that is converted intracellularly to cidofovir and its active metabolites. The half-life of brincidofovir is 19.3 h and of its metabolite cidofovir 113 h [30,99]. Of the parent drug, 51% is excreted as metabolites in urine and 40% as metabolites in feces. Drug transporters OATP1B1 and 1B3 are involved and concomitant use of inhibitors such as ciclosporin and erythromycin should be used with caution [30].

### 9.3. Pharmacodynamics

A case report of four cancer patients with resistant CMV or HSV infection that were treated with brincidofovir (100 or 200 mg, twice weekly (BIW)) under emergency IND application (early access) showed promising results [100]. However, in a placebo-controlled phase 3 trial of CMV-seropositive HCT recipients without CMV viremia at screening, brincidofovir 100 mg BIW prophylaxis did not reduce CMV viremia and/or disease necessitating antiviral therapy (CS-CMVi) by week 24 post-HCT and was associated with gastrointestinal toxicity [101]. A second clinical study showed only limited efficacy of BCV for HSV/VZV prophylaxis in HCT patients [102]. Recently there is renewed interest in brincidofovir due to monkeypox [98,103]. Unfortunately, the three monkeypox-infected patients that were treated with brincidofovir (200 mg QIW) developed elevated liver enzymes resulting in the cessation of therapy. This suggests that brincidofovir treatment in immune-deficient patients, when available, may have a place as second-line for CMV or HSV refractory infections. Brincidofovir treatment may result in an increased risk of liver toxicity, which is likely reversible upon cessation.

## 10. Optimized HSV and CMV Treatment Guidelines

Immunocompromised patients are vulnerable to HSV and CMV infections. Valacyclovir (500 mg BID PO) is the common antiviral agent for HSV prophylaxis in immunocompromised patients. Letermovir (480 mg QD PO or 240 mg QD PO with concomitant use of cyclosporine) can be added as CMV prophylaxis for CMV-positive patients transplanted with a CMV-negative donor, after lymphocyte depletion. Despite prophylactic therapy, HSV and CMV infections are common in immunocompromised patients and treatment options remain limited [104].

HSV infections or reactivations can clinically manifest on different anatomical sites (e.g.,: mucocutaneus, ocular, neurological (encephalitis), and respiratory tract). An HSV infection or reactivation in immunocompromised patients requires prompt treatment initiation [4]. First-line therapy is oral valacyclovir 1000 mg BID or intravenous acyclovir (5 mg/kg TID; dose depending on the anatomical site involved). We recommend TDM when a response is not observed within one week. Earlier TDM could be performed in case of other reasons such as decreased absorption or renal failure. When the target concentration is not reached, the dose can be increased linearly to reach the target concentration. Second-line systemic therapy (in case of acyclovir resistance or continued nonresponse) is foscarnet (40 mg/kg TIS IV or 90 mg/kg BID IV), combined with TDM in case of (renal) toxicity or delayed/nonresponse. The standard duration of the therapy is also dependent on the site of the disease and ranges for example from five to seven days for stomatitis to 10 to 14 days for encephalitis, but can be extended based on clinical manifestation. In case of nonresponse to foscarnet at adequate exposure or in case of nephrotoxicity, pritelivir or amenamevir are recommended.

In case of CMV reactivation (>1000 IU/mL EDTA plasma) valganciclovir (900 mg BID PO) is initiated. Intravenous administration of ganciclovir (5 mg/kg BID) can be considered in case of inability for oral administration or reduced enteral absorption (e.g., graft versus host disease (GvHD) of the gastrointestinal tract). The prophylactic treatment with valacyclovir will be stopped and immunosuppressive drugs (e.g., ciclosporin and tacrolimus) are reconsidered. We recommend TDM when a response is not observed within one week. Earlier TDM could be performed in case of other reasons such as a decreased absorption or renal failure. Treatment takes for at least two weeks or longer until the PCR result is <1000 IU/mL CMV disease is treated with ganciclovir (5 mg/kg BID IV) and switched after two weeks (if treatment response is sufficient) to valganciclovir (900 mg BID PO). This PO treatment continues for at least 2 weeks or longer based on PCR results and clinical state. When the first two weeks of IV ganciclovir does not result in a clear improvement, resistance is determined. Foscarnet is used as second-line treatment. However, in case of nephrotoxicity or nonresponse to foscarnet, a switch to brincidovir would be recommended—if the drug were available for the clinic.

Therapy-resistant CMV is defined as a 1 log10 increase of the PCR results in seven days or a less than 1 log10 decrease in the PCR results in 14 days with the same assay during treatment with a drug. Cessation of immunosuppressive drugs may be reconsidered or doses lowered. Furthermore, upon insufficient response after four treatment days, doses may be adjusted based on TDM: (val)ganciclovir trough concentrations should reach the target range of 2.0–4.0 mg/L (Table 1). Alternative therapies for (val)ganciclovir are foscarnet (90 mg/kg BID IV) or cidofovir (5 mg/kg QW IV; first 2 doses), combined with TDM in case of (renal) toxicity or delayed/nonresponse. Foscarnet can also be administered as a pre-emptive therapy in case of a contraindication for (val)ganciclovir (e.g.,: neurological toxicity or myelotoxicity). In case of nephrotoxicity or nonresponse, access to brincidovir is highly needed [105,106].

For antiviral treatment to be successful, the virus should be controlled by the immune system. However, this can be difficult in immunocompromised patients. To enhance the activity of the immune system, one may reconsider or lower the dose of immunosuppressive drugs in allo-HCT patients. This strategy may benefit the antiviral treatment but can lead to an increased risk of GvHD. In addition, the administration of intravenous immunoglobulin G (IVIG) is discussed frequently in literature [4,104,107]. No evident superiority compared to standard treatment has been demonstrated yet, and IVIG cannot replace antiviral treatment in acyclovir-resistant HSV infections in immunocompromised patients [104,107]. CMV-IVIG may benefit immunocompromised patients with CMV pneumonia [4]. Overall, IVIG seems safe, with limited side effects compared to the standard of care [104]. Since there is only limited evidence for the use of IVIG, we recommend careful patient selection when prescribing IVIG. Finally, virus-specific T cells (VSTs) could be, if available, an effective treatment in case of CMV infection [108]. There are no clinical studies with VSTs to HSV infections [109].

## 11. Discussion

This scoping review discusses optimized treatment guidelines for HSV and CMV infections based on pharmacokinetic (PK) and pharmacodynamic (PD) knowledge of antiviral drugs in immunocompromised patients. (Val)acyclovir and foscarnet form the cornerstone for HSV infections and (val)ganciclovir, foscarnet, and cidofovir for CMV infections. These antiviral drugs can be nephrotoxic. Moreover, the use of (val)acyclovir and (val)ganciclovir can be limited due to neurotoxicity, (val)ganciclovir can induce myelotoxicity and, foscarnet is associated with anemia and electrolyte disturbances. Treatment optimization is desirable to balance toxicity while overcoming the increasing risk of severe and treatment-resistant HSV and CMV infections during prolonged treatment. Here, we propose to optimize the treatment of HSV and CMV infections in immunocompromised patients based on early detection of drug resistance, optimization of dose to target concentration by therapeutic drug monitoring of nucleoside analogs, the introduction of new antiviral drugs such as brincidofovir, pritelivir, and amenamevir at an earlier stage, and a combination of synergistic antiviral drugs. The limited number of patients (and thus the limited revenue of new drugs), limited efficacy of current investigational products and the need of high dosages to achieve therapeutic concentrations in all viral target sites are possible limitations in the development of new antiviral drugs for immunocompromised patients.

Immunocompromised patients are more likely than immunocompetent patients to develop therapy-resistant HSV or CMV infections [110,111]. Early detection of acyclovir-resistant HSV infections or ganciclovir-resistant CMV infections to optimize treatment regimens would benefit these patients. It is recommended to determine drug resistance throughout treatment and not only in case of refractory disease. Drug resistance can be determined via genotypic and phenotypic testing. The fast turnaround of genotypic testing is an advantage, but numerous novel amino acid substitutions are diagnostically less conclusive [112]. Meanwhile, phenotypic testing offers a slower, but more complete result on drug resistance at lower viral loads compared to genotyping. In case of adequate drug exposure and increasing CMV/HSV load, we therefore, recommend performing a swab for genotyping drug resistance first. When CMV/HSV increases above 1000 copies/mL, we then recommend to consider phenotypic testing.

In this review, we advocate therapeutic drug monitoring (TDM) to optimize antiviral dosing. Effective application of TDM is bound to several conditions: (1) there is a narrow therapeutic window, (2) there should be a large interindividual variation in pharmacokinetics, and (3) a clear relationship between concentration, efficacy, and toxicity is established. Large interindividual variability in pharmacokinetics and pharmacodynamics has been observed for all nucleotide analogs, especially in immunocompromised/allo-HCT patients. There is a large risk of toxicity when overdosing and nonresponse, and resistance in case of underdosing of these drugs. Unfortunately, although the pharmacokinetics of some nucleotide analogs (e.g., acyclovir and ganciclovir) are well known, no clear therapeutic window has been defined yet. The non-nucleotide analogs, such as pritelivir, mabiravir and letermovir, have a more favorable toxicity profile. TDM could play a role in optimizing the efficacy of these drugs, most probable in patients with an abnormal volume of distribution or reduced clearance due to, e.g., organ failure, but more research is needed.

The current manuscript provides the first guidance for TDM-guided dosing. Its benefit should be verified in prospective randomized controlled intervention studies. In addition, there is a need to understand better the relationship between plasma total exposure and intracellular nucleoside triphosphates or other intracellular concentrations and subsequent inhibition of viral replication. To further improve our understanding of PK/PD of antivirals and optimize treatment in immunocompromised patients, the patient’s immune response should be quantified and exposure to different immunosuppressives or immune enhancers in relation to viral clearance should be assessed. The interferon-γ release assay for cytomegalovirus (IGRA-CMV) could be applied to quantify this immune response [113].

The presented cases in this review show that there is a need for more effective antiviral treatment options. Future studies should focus on new treatment strategies such as combination treatment of synergistic antiviral drugs (e.g., pritelivir/acyclovir), or alternation between nucleoside/non-nucleoside compounds to avoid toxicity and resistance. Furthermore, pharmaceutical companies should strive for a more rapid introduction of new compounds, emphasized by the not-yet commercialized brincidofovir twenty-six years after the first FDA/EMA approval. Existing HSV and CMV treatment in immunocompromised patients can be optimized using careful drug selection based on the drugs’ sensitivity and toxicity profiles, and doses can be optimized by TDM using validated plasma or target site concentrations. Therefore, future research should focus on randomized controlled trials to prospectively evaluate the benefit of TDM and optimal target concentration per compound.

## Figures and Tables

**Figure 1 pharmaceutics-15-00163-f001:**
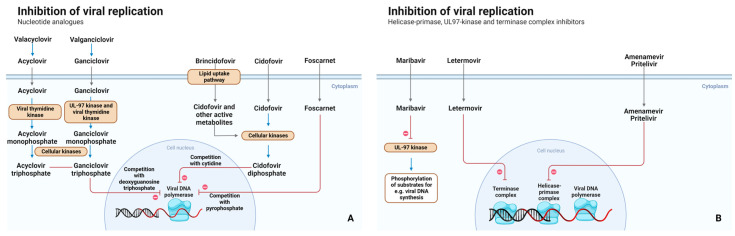
Mechanism of action of nucleotide analogs (**A**), helicase-primase, UL97-primase and terminase complex inhibitors (**B**).

## Data Availability

Data sharing not applicable.

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
