# Peer review of "Optimizing Antiviral Dosing for HSV and CMV Treatment in Immunocompromised Patients"

_pharmaceutics, 2023, doi:10.3390/pharmaceutics15010163_

Round 1

Reviewer 1 Report

Minor revisions

(A) In the introduction, add WHO information on the incidence of HSV and provide a division - herpes simplex virus (HSV) is categorized into 2 types: HSV-1 and HSV-2) and expand this division

(B) add information (www.cdc.gov/cmv/index.html) on CMV incidence in the introduction.

(C) Wrong cited literature (see: Instruction for Authors)

Is:

The reported incidence of viral 41 infections in allo-HSCT recipients is inconsistent; however, the incidence of HSV and 42 CMV infections has decreased in recent years due to effective preemptive therapies and prophylaxis. (1)

should:

The reported incidence of viral 41 infections in allo-HSCT recipients is inconsistent; however, the incidence of HSV and 42 CMV infections has decreased in recent years due to effective preventive therapies and prophylaxis [1].

Check and correct it all the text.

Author Response

We thank the reviewer for the feedback. 

According to the suggestions, we summarize shortly our revisions:

A) this is added to the manuscript

B) this is also added to the manuscript

C) we updated all references according to the reference style mentioned in the author guidelines.

Reviewer 2 Report

This review article aims to discuss the current information available on pharmacokinetic and pharmacodynamic data on antiviral therapeutics for HSV and CMV treatment in immunocompromised patients. It is a well-written review article which can be further improved with the following:

1) Abbreviations used for HSV and CMV should be consistently used across the manuscript. Eg. Line 175, the authors are still describing the virus as herpes simplex virus, though it has been termed as HSV in the earlier sections.

2) Illustrations and data in Figure 1 and Box Figure should be labelled as (A) and (B).

3) Authors should spellcheck acyclovir, brincidofovir and valacyclovir across the manuscript and tables.

4) Table 1: Authors should be consistent in placing uM within brackets.

5) Table 1: Any unavailable information should not be kept as blank spaces within the table. Authors should gather more information and summarized them in Table 1 from various literatures. Eg. EC50 for acyclovir is available in other  publications apart from the literature cited as (16).

6) What are the limitations on the developments of an optimally effective therapeutic options for HSV and CMV in immunocompromised patients thus far?

Author Response

We thank the reviewer for the feedback. 

According to the suggestions, we summarize shortly our revisions:

1) We edited the manuscript accordingly

2) The figures are separated

3) We corrected the drug names throughout the manuscript (including figures)

4) We edited the table accordingly

5) We clarified this in table 1. 

6) We added a small section to the discussion (line 764-768)

Reviewer 3 Report

148 please increase the font size (by a considerable amount) in figure 1 so it can be read

203 replace EC50 with EC50 please do this throughout the manuscript

225 Table 1- please make sure all the “50” are font effect subscript

231-232 table 1- please try to make all numerical values are on the same line, correct the symbols and exchange u for µ and 50 is font effect subscript. Please add dashes to empty boxes. Font should be reduced so all values are on the same line if possible

304 replace IC50 with IC50

Author Response

We thank the reviewer for the feedback. 

According to the suggestions, we summarize shortly our revisions:

148) we changed the fonts in the figure to make the figure more readable

203) the manuscript is edited accordingly

231) the table is edited accordingly

304) the manuscript is edited accordingly